# Rhabdoviral Endogenous Sequences Identified in the Leishmaniasis Vector *Lutzomyia longipalpis* Are Widespread in Sandflies from South America

**DOI:** 10.3390/v16030395

**Published:** 2024-03-02

**Authors:** Antonio J. Tempone, Monique de Souza Zezza-Ramalho, Daniel Borely, André N. Pitaluga, Reginaldo Peçanha Brazil, Sinval P. Brandão-Filho, Felipe A. C. Pessoa, Rafaela V. Bruno, Filipe A. Carvalho-Costa, Oscar D. Salomón, Petr Volf, Barbara A. Burleigh, Eric R. G. R. Aguiar, Yara M. Traub-Cseko

**Affiliations:** 1Laboratório de Biologia Molecular de Parasitas e Vetores, Instituto Oswaldo Cruz, Fiocruz, Rio de Janeiro 21040-360, RJ, Brazil; moniquezezza@gmail.com (M.d.S.Z.-R.); danielborely@hotmail.com (D.B.); pitaluga@ioc.fiocruz.br (A.N.P.); 2Instituto Nacional de Ciência e Tecnologia em Entomologia Molecular/CNPq, Rio de Janeiro 21040-360, RJ, Brazil; rafaelav@ioc.fiocruz.br; 3Laboratório de Doenças Parasitárias, Instituto Oswaldo Cruz, Fiocruz, Rio de Janeiro 21040-360, RJ, Brazil; brazil.reginaldo@gmail.com; 4Departamento de Imunologia, Instituto Aggeu Magalhães, Fiocruz, Recife 50740-465, PE, Brazil; sinval.brandao@fiocruz.br; 5Laboratório de Ecologia de Doenças Transmissíveis na Amazônia, Instituto Leônidas e Maria Deane, Fiocruz Amazônia, Manaus 69027-070, AM, Brazil; felipe.pessoa@fiocruz.br; 6Laboratório de Biologia Molecular de Insetos, Instituto Oswaldo Cruz, Fiocruz, Rio de Janeiro 21040-360, RJ, Brazil; 7Laboratório de Epidemiologia e Sistemática Molecular, Instituto Oswaldo Cruz, Fiocruz, Rio de Janeiro 21040-360, RJ, Brazil; carvalhocosta70@hotmail.com; 8Instituto Nacional de Medicina Tropical, Ministerio de Salud de la Nación, ANLIS, Puerto Iguazu 3370, Misiones, Argentina; dsalomon@msal.gov.ar; 9Department of Parasitology, Charles University, 12800 Prague, Czech Republic; volf@cesnet.cz; 10Department of Immunology and Infectious Diseases, Harvard School of Public Health, Cambridge, MA 02115, USA; bburleig@hsph.harvard.edu; 11Departamento de Bioquímica e Imunologia, Instituto de Ciências Biológicas, Universidade Federal de Minas Gerais, Belo Horizonte 31270-901, MG, Brazil; ericgdp@gmail.com

**Keywords:** *Lutzomyia longipalpis*, endogenous viral element, PIWI-RNA

## Abstract

Sandflies are known vectors of leishmaniasis. In the Old World, sandflies are also vectors of viruses while little is known about the capacity of New World insects to transmit viruses to humans. Here, we relate the identification of RNA sequences with homology to rhabdovirus nucleocapsids (NcPs) genes, initially in the *Lutzomyia longipalpis* LL5 cell lineage, named NcP1.1 and NcP2. The Rhabdoviridae family never retrotranscribes its RNA genome to DNA. The sequences here described were identified in cDNA and DNA from LL-5 cells and in adult insects indicating that they are transcribed endogenous viral elements (EVEs). The presence of NcP1.1 and NcP2 in the *L. longipalpis* genome was confirmed in silico. In addition to showing the genomic location of NcP1.1 and NcP2, we identified another rhabdoviral insertion named NcP1.2. Analysis of small RNA molecules derived from these sequences showed that NcP1.1 and NcP1.2 present a profile consistent with elements targeted by primary piRNAs, while NcP2 was restricted to the degradation profile. The presence of NcP1.1 and NcP2 was investigated in sandfly populations from South America and the Old World. These EVEs are shared by different sandfly populations in South America while none of the Old World species studied presented the insertions.

## 1. Introduction

Leishmaniasis, caused by parasites of the genus *Leishmania*, is a serious public health problem. Parasites are transmitted through the bite of infected sandflies and can cause different disease manifestations. Among these visceral leishmaniasis (VL) diseases is the most severe form of the disease that can lead to death in untreated cases [1]. In the New World, VL is mostly caused by *Leishmania infantum* [2], which is transmitted by the sandfly *Lutzomyia longipalpis*. *Lutzomyia*, the most important New World sandfly genus in terms of species diversity and medical importance exhibits a wide distribution area covering South and Central Americas [3,4,5].

Sandflies are mostly known for transmitting leishmaniasis but can also harbor and transmit viruses [6]. Although in Europe, these insects of the genus *Phebotomus* represent important viral vectors [7], little is known about sandfly-borne viruses in the Americas. One example is the vesicular stomatitis virus, which infects humans and domestic animals and is widely endemic in the New World [8]. There are reports on the isolation of arboviruses from sandflies in several areas of the Amazon [9,10]. In sandflies collected from Brazil, Colombia and Guatemala, five different new phleboviruses were isolated [11]. In 2020, through in silico analyses and RT-PCR experiments, a mitovirus was identified in *L. longipalpis* [12].

One of the consequences of viral infections is the possibility of partial or total integration of the viral genetic material into the host genome. Until the last decade, it was believed that this ability was exclusive to retroviruses. Retroviruses have a genome made up of positive single-stranded RNA, which, during the infectious process, retrotranscribes its genome into complementary DNA (cDNA), which, through an integrase enzyme, is incorporated into the genome of infected cells. These retroviral sequences inserted into the host genome are known as endogenous retroviruses (ERVs). ERVs are very common in vertebrate genomes, making up about 8% of the human genome [13]. In 2010, Horie et al. identified viral elements of non-retroviral origin in the genome of several mammals. After that, the integration of different viruses into the genome of several eukaryotes, including invertebrates and plants, was shown [14,15,16,17,18]. In silico studies identified the presence of endogenous viral elements (EVEs) in arthropods used as a model organism or of medical interest. These studies were carried out on the mosquitoes *Aedes aegypti*, *Culex quinquefasciatus*, the tick *Ixodes scapularis* and the sandflies *L. longipalpis*, *Phlebotomus duboscqi* and *Sergentomyia* sp. Showing the occurrence of fragments of rhabdovirus and other viruses integrated into the genomes [19,20]. In a study in which the genomes of 48 different arthropods were analyzed, researchers observed that EVEs were widely present in these organisms. The majority came from the integration of unclassified single-stranded RNA (ssRNA) viruses and viruses from the *Rhabdoviradae* and *Parvoviridae* families, and most of these EVEs were located in piRNA clusters transcribing piRNAs [21]. The EVEs can be considered fossils of viral infections that occurred in the past, which somehow remained in the genome of their hosts and descendants over time. The integration of exogenous viral sequences in the genome can be deleterious, neutral, or positive to hosts. The viral sequences integrations in the germ line cells with deleterious effect to the host may be lost in a single host generation and only slight deleterious, neutral or advantageous insertions may expand to the host population. In this way, EVEs can be used as an element of phylogenetic analysis between different populations [19]. Some of these EVEs that have even been conserved and selected over long evolutionary timescales can confer new essential functions to their hosts. This phenomenon is known as exaptation, when a molecule or structure evolves to a different function than it was originally designed for. Increasing evidence in insects shows that most EVEs are transcriptionally active and produce small interfering RNA sequences (sRNAs) [22]. Although the exact function of EVEs is so far unknown, some studies suggest that EVEs may interfere with virus replication by producing PIWI-interacting RNA sequences (piRNAs) that recognize and degrade viral RNA sequences through sequence complementarity [23,24]. The exaptative process involving EVEs is exemplified in these cases by the change in the function of sequences that were previously related to the expression of viral proteins and that have now become part of the host’s immune system.

In the present work, we detected two viral RNA sequences coding for rhabdoviral nucleocapsid proteins in *L. longipalpis* LL5 cells’ exosomes and in adults from our colony formed by insects collected in Jacobina, BA, Brazil. Posteriorly, we determined that these viral sequences were not derived from virus infection but were the product of the transcription of viral elements inserted in intron regions of two putative protein-coding genes in the *L. longipalpis* genome, being classified as endogenous viral elements (EVEs). These EVEs were named NcP1.1 and NcP2. A subsequent in silico analysis showed the existence of a third viral insertion, located in the same intronic region where NcP1.1 was located. This new EVE was called NcP1.2. Pre-processing of the small RNA library revealed that NcP1.1 and NcP1.2 presented a profile consistent with elements targeted by primary piRNAs and therefore could play a role in sandfly immunity. This profile was not identified for NcP2. We also investigated whether other sandfly populations and species in the New and Old Worlds shared these EVEs. We observed that diverse sandflies from different regions of South America shared these EVEs. None of the *Phlebotomus* species from the Old World studied here showed such EVEs.

## 2. Methods

### 2.1. Sandflies

*L. longipalpis* from our colony, originally collected in Jacobina (Bahia, Brazil), were kept in our insectary at 26 °C and fed on 70% sucrose solution ad libitum. Females were fed on anesthetized hamsters when needed. Sandflies collected in the field were immediately added to a TRIzol^®^ reagent (Invitrogen^®^, Carlsbad, CA, USA) for processing. Sandflies from the Old World were kept as described in [25].

### 2.2. Lutzomyia Longipalpis Cell Lines

LL-5 and embryonic cells originally isolated from *L. longipalpis* from Lapinha, MG, BR, were maintained in L-15 medium (Sigma-Aldrich Co., Spruce Street, St. Louis, MO, USA) supplemented with 10% fetal bovine serum (FBS) (Laborclin, Pinhais, Brazil) and 10% tryptose phosphate broth (TPB) at 28 °C.

### 2.3. RNA and DNA Extractions

RNA and DNA were extracted using the TRIzol^®^ reagent (Invitrogen^®^, Carlsbad, CA, USA), according to the manufacturer’s instructions. RNA was stored at −80 °C and DNA at −20 °C.

### 2.4. cDNA Synthesis

After RNA extraction, a possible contamination with genomic DNA was verified and, when present, the samples were treated with the RNA-free DNAse TURBO DNA-free Kit (Ambion, Austin, TX, USA). cDNA was synthesized using the SuperScript^®^ III First-StrandSynthesis System kit (Invitrogen^®^, Carlsbad, CA, USA), according to the manufacturer’s recommendations.

### 2.5. Polymerase Chain Reactions (PCR)

Primers (Table 1) were designed using the Oligonucleotide-BLAST (NCBI-NIH) and AmplifX programs, available at: https://www.ncbi.nlm.nih.gov/tools/primer-blast/ (accessed on 30 March 2023) and https://amplifx.Software.informer.com/1.7/ (accessed on 30 March 2023), respectively. The detection of viral sequences in cDNA samples was performed by conventional PCR reactions using the following conditions: initial denaturation at 95 °C for 5 min, followed by 35 cycles of denaturation at 95 °C for 30 s, annealing at 56 °C for 30 s and extension at 72 °C for 30 s, followed by an additional extension step of 5 min at 72 °C. The DNA reactions were conducted with the following temperature conditions: initial denaturation at 95 °C for 5 min followed by 35 cycles of denaturation at 95 °C for 45 s, annealing at 57 °C for 45 s and extension at 72 °C for 1 min and 30 s and then an additional extension step of 5 min at 72 °C.

### 2.6. Agarose Gel Electrophoresis

Samples amplified by PCR were submitted to electrophoresis in 1.5% agarose gels in Tris-acetate-EDTA (TAE) buffer at 1× concentration containing 0.5 µg/mL of ethidium bromide at 110 mV.

### 2.7. Bioinformatics Tools

The genomic insertions identified in *L. longipalpis* were investigated in silico using the BLAST tool (Basic Local Alignment Search Tool—https://blast.ncbi.nlm.nih.gov/Blast.cgi v. 2.13.0) with the NCBI databases, VIPR (VirusPathogenResearch—https://www.viprbrc.org/brc/home.spg?decorator=vipr v. 3.34.11) and VectorBase (https://vectorbase.org/vectorbase/app release 67). (https://www.ncbi.nlm.nih.gov/tools/vecscreen/ 22-May-2017 build 10.0). Multiple sequence alignments, visualization, and analyses were performed using the Jailview program (http://www.jalview.org/getdown/release/ v. 2.11.3.2).

### 2.8. Alignment

The nucleotide sequences of EVEs NcP1.1, NcP1.2 and NcP2 were submitted as probe for homology searches using the BlastX tool (Basic Local Alignment Search Tool—https://blast.ncbi.nlm.nih.gov/Blast.cgi v. 2.13.0) against the NCBI databases (National Center for Biotechnology Information (Bethesda, MD, USA)) The EVE-deduced amino sequences obtained in BlastX were aligned with nucleocapsid proteins from rhabdovirus using the ClustalW v. 2.1 multiple alignment tool [26].

### 2.9. Evolutionary Analysis by Maximum Likelihood Method

The evolutionary history was inferred by using the maximum likelihood method with a JTT matrix-based model [27]. The tree with the highest log likelihood (−1932.07) is shown. Initial tree(s) for the heuristic search were obtained automatically by applying the neighbor-join and BioNJ algorithms to a matrix of pairwise distances estimated using the JTT model and then selecting the topology with superior log likelihood value. The tree was drawn to scale, with branch lengths measured in the number of substitutions per site. This analysis involved 11 amino acid sequences. There was a total of 96 positions in the final dataset. Evolutionary analyses were conducted using MEGA11 v. 11.0.13 [28].

### 2.10. Small RNA Analysis

Public *L. longipalpis* small RNA libraries were downloaded from NCBI SRA database and the reads merged into one single file to increase depth. Pre-processing of the resultant RNA library was performed as described [29]. Briefly, raw sequences were submitted to quality filters and adaptor removal. Sequences with low Phred quality (<20), ambiguous nucleotides and/or a length shorter than 15 nt were eliminated. Pre-processed reads were aligned against reference sequences using the Bowtie program (v1.1) [30] accepting 1 mismatch. The putative Rhabdovirus sequences were compared against the reference genome of the *L. longipalpis* reference genome, (Jacobina strain, version J1.2) downloaded from the VectorBase website (www.vectorbase.com, accessed on 14 April 2022) using BLAST software v. 2.12.0 [31] in its BlastN variation requiring e-value < 1 × 10^−5^. The analysis of the small RNA size profile, 5′ base preference, density of coverage, and additional data analysis were evaluated using in-house Perl and R scripts.

## 3. Results

### 3.1. Determination of the Origin of RNA Sequences Coding for Viral Proteins in Exosomal Fraction of LL5 Cells

In previous work, nucleic acid sequencing was performed on an exosomal pellet from *L. longipalpis* LL5 embryonic cells. After alignments with databases, two partial RNA coding sequences showing similarity to rhabdovirus nucleocapsid proteins with 477 and 459 nucleotides were identified and named NcP1.1 and NcP2, respectively (Appendix A).

The presence of RNA fragments coding for rhabdoviral nucleocapsid proteins in the exosomal fraction of LL5 cells raised the question regarding their origin. These sequences could be derived from an exogenous viral infection or from viral insertions in the insect genome. Since members of the *Rhabdoviridae* family are negative single-stranded RNA viruses, whose genome is never in the form of DNA, we performed PCR assays, using DNA and cDNA samples from LL5 cells as templates to answer this question (Figure 1).

Both templates were positive, revealing that the sequences were derived from the transcription of viral elements inserted in the genome of LL5 cells. These results were confirmed by the analysis of the *L. longipalpis* genome data deposited on the Vector Base and NCBI database sites using the BlastN tool v. 2.13.0.

### 3.2. Genomic Context of NcP1.1 and NcP2 in the L. longipalpis Genome

BlastN using the NcP1.1 and NcP2 sequences as bait against the Vector Base *L. longipalpis* deposited genome confirmed the result obtained with the PCR experiments, showing that both sequences were present in the genome and located in intronic regions of two deduced protein-coding genes. The EVE NcP1.1 is located between the nucleotides 63,562 and 64,974 of the supercontig JH689452. This region is in the intron of the unannotated putative protein-coding gene LLOJ001560. Surprisingly, the BlastP analysis of the deduced amino acid sequence encoded by this gene against the NCBI data bank showed homology and 35% of identity with the viral capsid protein from Nebet virus Seq ID: QRW425091. Furthermore, the BlastX tool analysis of the region where NcP1.1 is located revealed the existence of another EVE with 950 bp, with homology to viral nucleocapsid protein, in the intron of this putative protein-coding gene, localized between nucleotides 62,442 and 61,493 of the supercontig JH689452. This new EVE was named NcP1.2. Interestingly, NcP1.1 and NcP1.2 present different transcriptional orientations (Figure 2). NcP2 was located between the nucleotides 58,124 and 59,430 of the supercontig JH689584, an intronic region of the deduced protein-coding gene LLOJ004474. The BlastP analysis of the deduced amino acid sequence encoded by this gene revealed homology with the hrp65 protein, which is related to the transport of RNA from the nucleus to the cytoplasm of the cell (Figure 2). In addition to confirming the result obtained with the PCR assays and the identification of another viral insertion in the genome (Ncp1.2), the analysis using the BlastN and BlastX tools against the *L. longipalpis* genome deposited on the Vector Base website revealed that the sequences NcP1.1 and NcP2 were larger than previously identified. Ncp1.1 and NcP2 went from 477 and 459 bp to 1413 and 1307 bp, respectively (Appendix A).

The BlastN analysis using the sequences of NcP1.1, NcP1.2 and NcP2 against the nucleotide database deposited at NCBI showed that NcP1.1 and NcP1.2 presented 100% of identity with the uncharacterized mRNA LOC129786293 and NcP2 showed 99% identity with four gaps in 1311 nucleotides, with the *L. longipalpis* hrp65 protein (LOC129793427), transcript variant X2 mRNA. This same analysis revealed that all three EVEs were located on chromosome I of *L. longipalpis* isolate SR_M1_2022. These results confirm what we observed experimentally about NcP1.1 and NcP2 and reveal that the EVE NcP1.2 is also transcribed in *L. longipalpis.*

### 3.3. Multiple Alignment and Phylogenetic Analysis of EVE-Deduced Proteins sequences

The sequences of NcP1.1, Ncp1.2 and NcP2 were translated and aligned with nucleocapsid protein sequences from various rhabdoviruses. The alignment suggested all of them were sequences from different rhabdovirus infections since they aligned in the same region of the deduced proteins. A phylogenetic analysis comparing the EVEs NcP1.1, Ncp1.2 and NcP2 with nucleocapsid sequences from modern rhabdoviruses revealed that NcP1.1 and NcP2 were evolutionarily closer to each other than to nucleocapsid from current rhabdoviruses. NcP1.2 also showed little proximity to the other modern rhabdoviruses (Figure 3).

### 3.4. Molecular Characteristics of Small RNA Sequences Derived from Rhabdoviral Sequences

It has been shown that the small RNA profile works as a proxy to determine the origin of the viral sequence [29,30,31,32]. Pre-processing of the small RNA library was performed as described [29] to determine whether the sequences NcP1.1, Ncp1.2 and NcP2 had characteristics of viral elements inserted into the genome. Sequences NcP1.1 and NcP1.2 presented a profile consistent with elements targeted by primary piRNAs (accumulation of RNA sequences in between 24 and 29 nt derived for only one strand) while NcP2 was restricted to a degradation profile (low abundance of small RNA sequences of different lengths). In addition, the density of small RNA sequences along the sequences were discontinuous, with hotspots in specific regions and coverage concentrated in one strand (Figure 4). Thus, since they presented most of the canonical features presented by EVEs in insects, they could be classified as endogenous elements.

### 3.5. Distribution of Rhabdoviral Sequences in Sandfly Populations

We investigated whether the EVEs NcP1.1 and NcP2 and their transcripts, identified in the genome of *L. longipalpis* from Jacobina, BA and LL5 cells from *L. longipalpis* from Lapinha, MG, BR, are found in different populations of sandflies from the New and Old Word.

A total of 61 insect samples, 58 from South America (53 from Brazil, 2 from Argentina and 1 from Colombia) were investigated (Figure 5). Five samples of different species from the Old World were obtained from insectary colonies. Either DNA or RNA, or both were extracted from these samples. Samples not analyzed were noted as not determined (ND). We observed that the EVEs NcP1.1 and NcP2 had a wide distribution, being present and transcribed by sandflies populations from all regions studied (Figure 6). We also found that Old World insectary specimens from the genus *Phlebotomous*, *Phlebotomous arabicus*, *Phlebotomous argentipes*, *P. papatasi*, *Phlebotomous sergenti* and *Phlebotomous schwetzi*, found in nature in Africa, Asia and Europe, did not present these EVEs in their genomes.

## 4. Discussion

The discovery of two RNA sequences coding for rhabdoviral nucleocapsid proteins present in the exosomal fraction of the *L. longipalpis* cell line LL5 and PCR assays using DNA and cDNA from this cell line as templates led us to search for these EVEs in the *L. longipalpis* genome. In silico analyses confirmed the presence of the sequences in the genome and revealed the existence of a third viral insertion.

Most EVE sequences so far identified were related to rhabdoviral elements. These seem to be widespread in eukaryotic genomes, since these rhabdoviral elements can be found in the genomes of many plants, insects and mammals [19,33,34,35,36]. More than 180 rhabdovirus sequences were identified in the genome of different animals, mainly in *A. aegypti* and *I. scapularis* [19], coding for nucleoproteins, glycoproteins and RNA-dependent RNA polymerases. These EVEs were acquired through evolution and serve as a record of previous viral infections. In silico studies showed that the *L. longipalpis* EVEs here identified were inserted into intronic regions of two putative protein-coding genes. The genes LLOJ004474, which codify for a homologous of hrp65 protein and the gene LLOJ001560, which, despite having a eukaryotic gene structure, putatively codifies for a protein with homology to a viral capsid. Intronic regions of protein-coding genes are transcribed together with exons and are subsequently edited and removed from mature mRNAs. The insertion of EVEs in these regions might be responsible for their transcription. The fact that the sequences described here are transcribed suggests an exaptative process, as reported [19].

The role of EVEs has been the subject of speculation and most evidence points to their participation in defense mechanisms. Their possible involvement in antiviral immunity has been shown for *A. aegypti* [37], where the presence of viral sequences integrated into the host genome limited the replication of the cognate virus. Recent evidence indicates that EVEs can function as models for the biogenesis of RNAs that interact with PIWI (piRNAs), P-element-induced wimpy in *Drosophila*. PIWI proteins are highly conserved RNA-binding proteins belonging to the Argonaute/PIWI family involved in RNA interference mechanisms present in many organisms [38]. The participation of piRNAs derived from EVEs in the innate antiviral immune response in mosquitoes have been demonstrated [39,40].

In our case, the analysis of small RNA sequences revealed that the derived small RNA profile from EVEs NcP1.1 and Ncp1.2 was similar to those observed for endogenous elements in different mosquitoes, such as *A. aegypti*, *A. albopictus* and *C. quinquefasciatus* [30,34]. They presented a profile consistent with elements targeted by primary piRNAs and a disform coverage. However, differently from *Aedes* mosquitoes that present a strong 5′ U base enrichment, *L. longipalpis* EVE-derived small RNA sequences did not show such enrichment. The lack of enrichment could be a specificity of EVEs in this vector or a common profile for *L. longipalpis* small RNAs, since the 5′ base enrichment was also absent in other virus-derived sequences [41]. Interestingly, the two EVEs identified as possible sources of piRNA production in *L. longipalpis* are located in an intron of a putative gene that codes for a protein with homology to a viral capsid protein. More studies will be needed to determine if this region of the *L. longipalpis* genome may be a mini-piRNA cluster. NcP2 transcripts were restricted to the degradation profile, presenting a low abundance of small RNA sequences of different lengths. An important observation regarding the EVEs studied in this work is that, as seen in Figure 3, they are substantially divergent of any possible cognate exogeneous virus, casting doubts about the role of these elements in the regulation of cognate viruses’ replication through sequence-dependent sRNA pathways. On the other hand, the identification of transcripts originating from the EVEs NcP1.1 and NcP2 in the exosomes of *L. longipalpis* LL5 cells is quite intriguing, given that, exosomes are subcellular vesicles of endosomal origin that participate in cellular communication processes. Exosomes from *L. longipalpis* LL5 cells were previously identified as involved in a nonspecific interferon-like antiviral response from these cells [42].

None of the Old World representatives investigated presented these EVEs. In contrast, we identified NcP1.1 and NcP2 insertions and transcriptions in many sandfly populations of South America and observed an impressive widespread presence of these sequences across geographical and genetic boundaries. This fact suggests that the event or events that led to the incorporation of these viral elements into the genome was exclusive to New World sandflies. Among the sixty-one samples, thirty-four were positive for NcP1.1 and eleven for NcP2, nine of which presented both EVEs, twenty-four only had EVE NcP1.1, and three only NCP2. Regarding the transcription of these EVEs, we observed that twelve transcribed NcP1.1, eleven transcribed NcP2, and nine transcribed both. Three only transcribed NcP1.1 and two only transcribed NcP2. Populations of *L. longipalpis* from Jacobina, BA, Lapinha, MG, Serra da Tiririca, RJ and Bogotá, COL, and *Psychodopygus davisi* from four different cities from Rondônia State carried and transcribed both EVEs. On the other hand, populations that do not present any of these insertions were also identified. With the exception of *Nyssomyia umbratilis* from Manacapuru, which had NcP1.1, but did not transcribe it, none of the sandflies detected in Amazonas presented NcP1.1 or NcP2. Populations of *L. longipalpis* as distant as Jacobina in northeastern Brazil and Bogotá in Colombia, approximately 4000 km apart in a straight line, and *P. davisi*, a sandfly species population of Rondônia State, separated approximately by 1800 km from Bogotá and 2600 km from Jacobina, share and transcribe both EVEs. Interestingly, *L. longipalis* captured in Bogotá, Colombia possesses and transcribes both EVEs, a fact that does not occur in sandflies from Amazonas but occurs with insects from Rondônia, which is further from Bogotá than Manaus. Another interesting aspect is the fact that the majority of specimens from the states of Amazonas and Pernambuco in the north and northeast of Brazil, respectively, do not present these EVEs. Thus, regarding the presence of NcP1.1, the sandflies from Amazonas, in the northern region of Brazil, are closer to the insects from the State of Pernambuco, located in the northeast of Brazil, than to the sandflies from the State of Rondônia, also located in the north of Brazil and much closer to Amazonas. A higher genetic relation proximity between sandfly populations in the states of Amazonas and Pernambuco was also observed in other sandfly population genetics studies using other markers [43,44].

All this indicates an ancient viral insertion event was transmitted and maintained throughout evolution. The fact that different species of sandflies from South America and none of the Old World species possess the EVEs NcP1.1 and NcP2 suggest that the infections that led to the introduction of these viral sequences into their respective genomes probably affected the germ cells of sandflies from South America before the speciation processes that led to the existence of some of the sandflies species. Mechanisms related to selective environmental pressures could be responsible for the maintenance or elimination of these viral sequences in the sandflies of the same species but from different populations.

Determining the real role of these EVEs in the biology of sand flies demands new and in-depth studies.

## Figures and Tables

**Figure 1 viruses-16-00395-f001:**
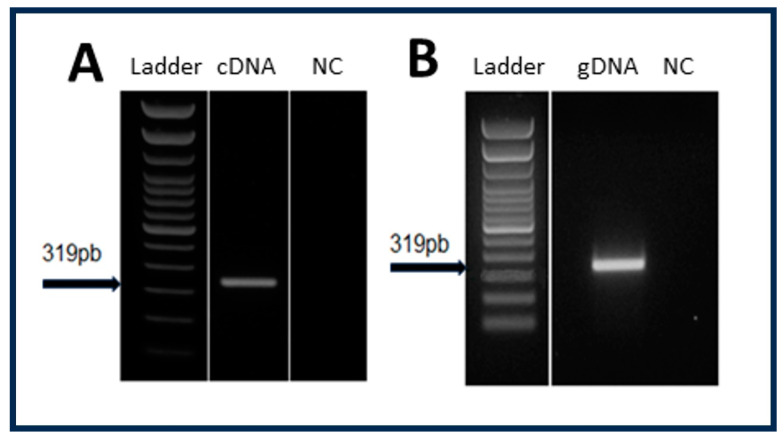
PCR assays to amplify the NcP1.1 sequence using cDNA (**A**) and DNA (**B**) from LL5 cell line as templates.

**Figure 2 viruses-16-00395-f002:**
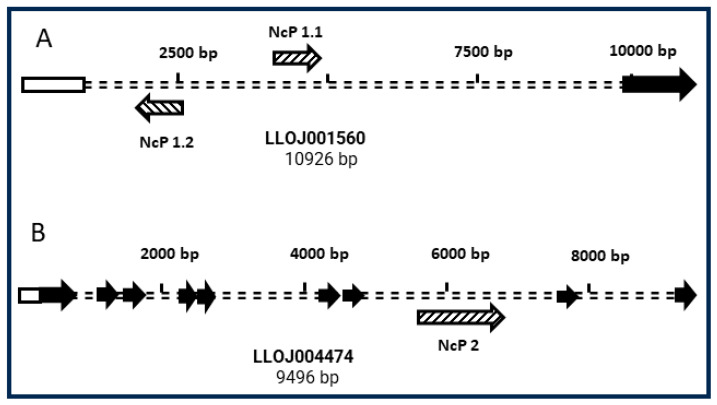
Graphical representation of the insertion sites and transcriptional sense of the EVEs NcP1.1, NcP1.2 and NcP2 in introns of *L. longipalpis* genes. (**A**)—Localization and sense of transcription of EVEs NcP1.1 and NcP1.2 (hatched arrows) in the intron of gene LLOJ001560 (double-dashed line). (**B**)—Localization and sense of transcription of EVE NcP2 (hatched arrow) in the intron of gene LLOJ004474 (double-dashed line). The EVEs’ transcription orientation is represented by the hatched arrows orientation. Black arrows represent the gene exons.

**Figure 3 viruses-16-00395-f003:**
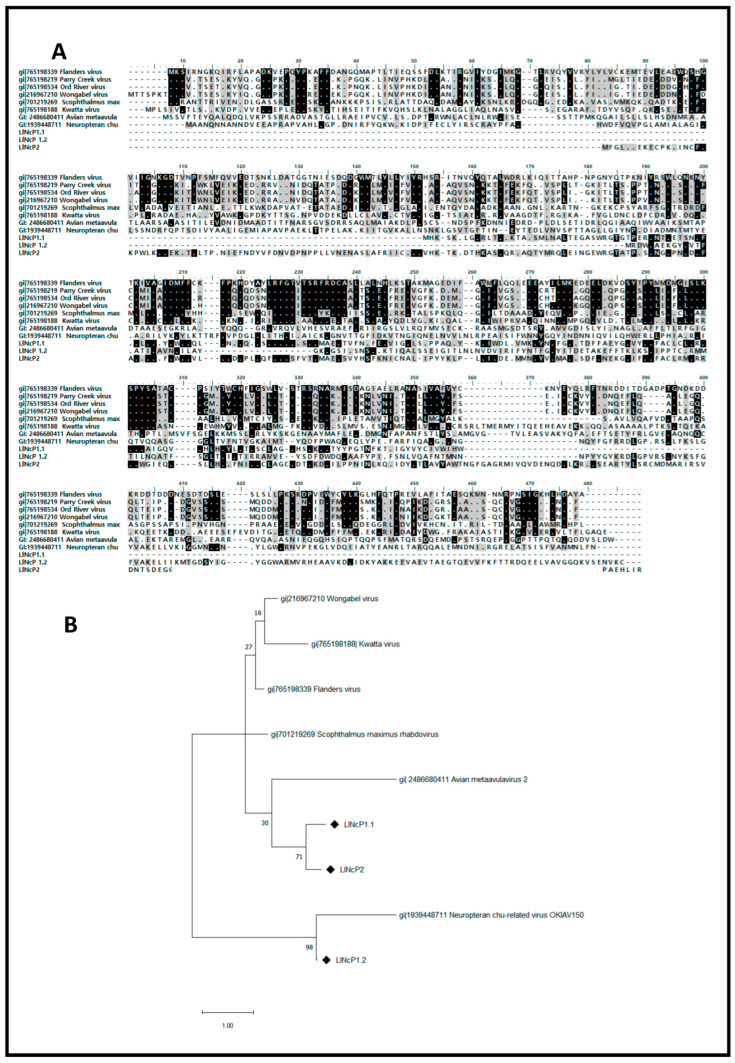
(**A**)—Alignment of deduced amino acids sequences of NcP1.1, NcP1.2 and NcP2 with other rhabdovirus nucleocapsid proteins. Regions with a gray or black background indicate similar or identical amino acids, respectively. (**B**)—Phylogenetic tree showing the relationship among these EVE-deduced (black rhombus) amino acid sequences and nucleocapsid protein sequences from rhabdoviruses.

**Figure 4 viruses-16-00395-f004:**
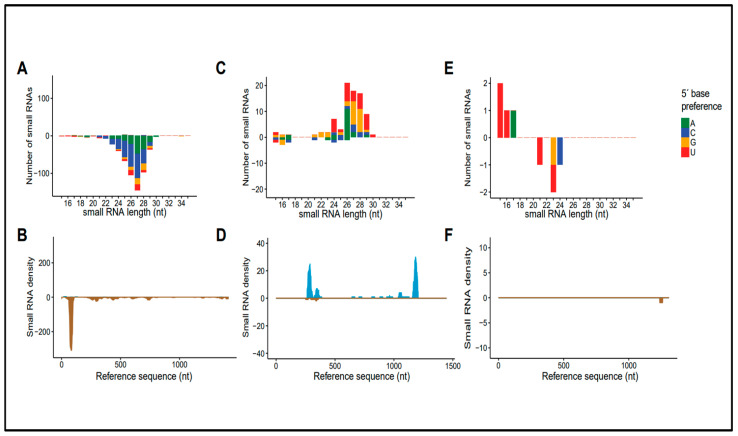
Small RNA profile derived from rhabdoviral endogenous sequences. Size distribution and coverage of small RNA sequences (24–29 nt) derived from NcP1.1 (**A**,**B**), NcP1.2 (**C**,**D**) and NcP2 (**E**,**F**).

**Figure 5 viruses-16-00395-f005:**
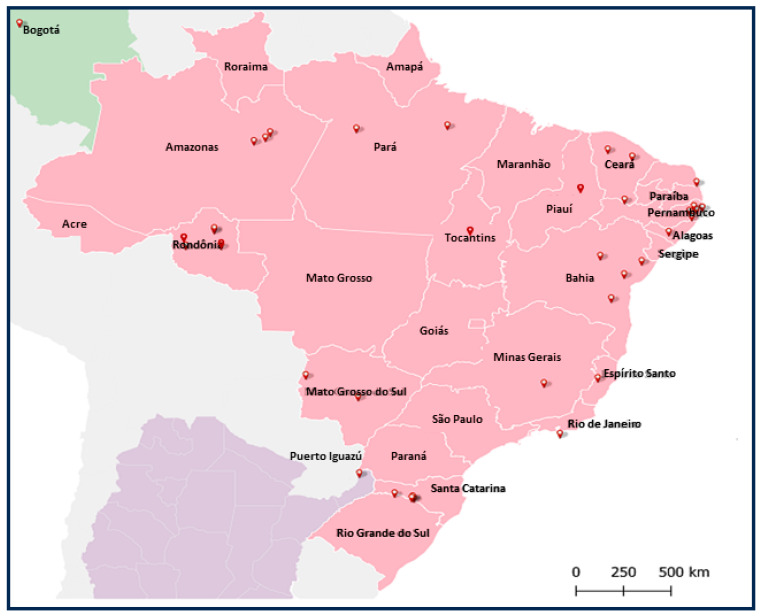
Geographic distribution of the origin of sandfly populations from South America analyzed in this work.

**Figure 6 viruses-16-00395-f006:**
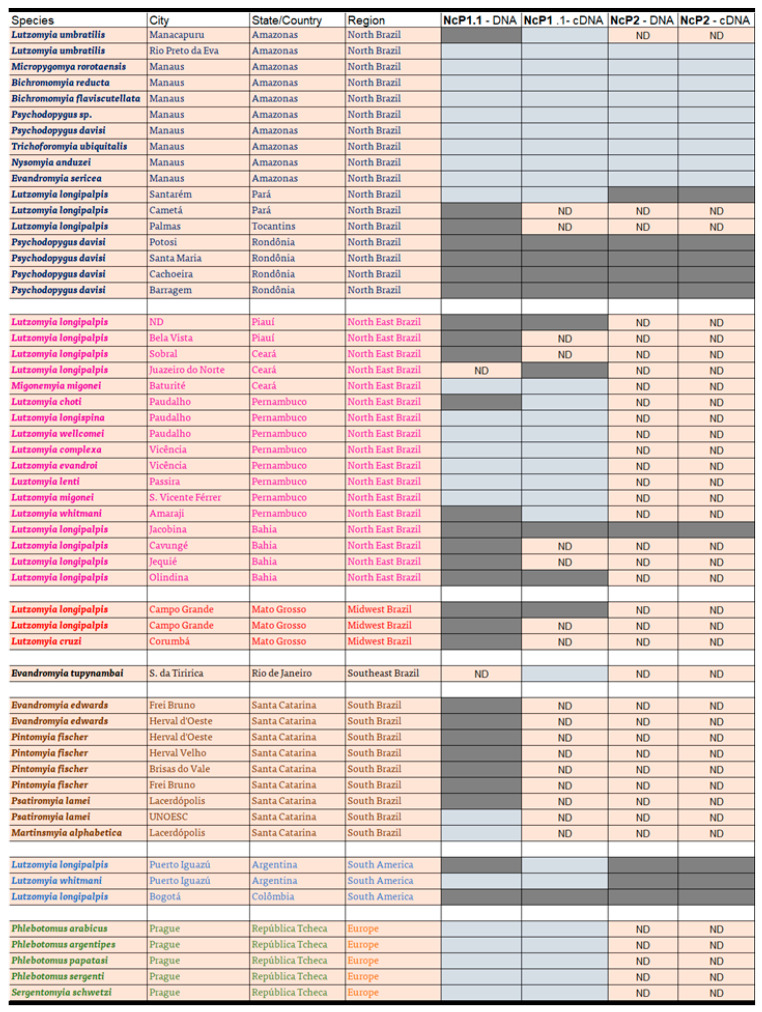
Determination of the presence and transcription of EVEs NcP1.1 and Ncp2 in sandflies from different populations of South America and the Old World. Green dark grey represents a positive result, while light gray indicates a negative result. ND means the experiment was not performed.

**Table 1 viruses-16-00395-t001:** Primers employed in PCR assays.

Name	Sequence	Amplicon Length (nt)
Histone forward	5′ GAAAAGCAGGCAAACACTC 3′	500 pb
Histone reverse	5′ GAAGGATGGGTGGAAAGG 3′	
NcP1.1 forward	5′ GGAACCCCAGAACGATACAA 3′	319 pb
NcP1.1 reverse	5′ GACAGAGGCACGCGAAGTAT 3′	
NcP2 forward	5′ TTCGAAGTGTCGCTTGCAGCC3′	415 pb
NcP2 reverse	5′ GCAACCCCAAACTCCTACAA 3′	

## Data Availability

Dataset available on request from the authors.

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
