# Peer review of "Rhabdoviral Endogenous Sequences Identified in the Leishmaniasis Vector Lutzomyia longipalpis Are Widespread in Sandflies from South America"

_viruses, 2024, doi:10.3390/v16030395_

Round 1

Reviewer 1 Report

Comments and Suggestions for Authors

The manuscript by Tempone et al. describes a comprehensive study of the endogenous viral elements (EVEs) identified in sandflies from South America. The nucleocapsid protein encoded sequences similar to rhabdovirus nucleoprotein gene were initially found in the genome of L. longipalpis embryonic cells, confirmed to reside in the genomic DNA of the insect, and later found in a specific chromosome locus of the L. longipalpis genome. Subsequent bioinformatics analysis confirmed the wide presence of similar NC sequences in genomes of other sandflies from S. America but not from the Old World. This is a very interesting and novel finding of a new type of EVEs in insect genomes that deserves to be published in ‘Viruses.” The manuscript is generally well-written, although may need some minor editing, since it visibly contains segments of text coming from different co-authors (even having different font types and sizes).

The manuscript is recommended for publication once minor issues listed below are properly addressed.   

Specific points to address:

l. 32 – Provide the full latinized name in this first instance, use the abbreviated name after that.

l. 43 – Same for the keyword line.

l. 82 – Correct ‘Rhabdoviridae’.

l. 102 – Should this be ‘two’ instead of ‘to’?

Comments on the Quality of English Language

Minor English editing may be needed.

Reviewer 2 Report

Comments and Suggestions for Authors

This paper reports three endogenous viral elements (EVEs) derived from a rhabdovirus nucelocapsid gene in the genome of sandfly Lutzomyia longipalpis.  Notably, these EVEs were present in some South American sandfly species, but were absent in the four sandfly species from Old world. 

Although such endogenous non-retroviral RNA virus elements in are relatively common in insect genomes, analysis of their distribution in related species (or populations of the same species) may provide a novel insight into evolution of both insects and viruses. Therefore I feel that analysis of the Rhabdovirus EVEs in sandflies warrants publishing in "Viruses" after addressing questions to MS, see below:

Major points. 

1. Complete detection of NcP2 (at least DNA detection), in particular in four samples from Europe/Prague and at least for one sample per species for South American.

2. Reorganize Table 6:

Sort according to insect species. i.e. all Lutzomyia longipalpis samples should be grouped together. This would make it easier to see if there are differences in the presence of EVE in different samples of the same species,  such as Lutzomyia longipalpis from Santarém and Jequié. 

In column "Region" add "Brazil" where required, e.g. " North" ->   " North Brazil" and so on. 

Replace red and grey fields (columns 5 to 8) with text, for example  - and + marks. to indicate for the absence or detection of EVEs. Current table is not suitable for color-blind people or for printing in black and white. 

3. Include a supplementary text with nucleotide sequences of the reported EVEs.

Other points: 

L 31-32 "with homology to Rhabdovirus Nucleocapsids.." -> "with homology to Rhabdovirus Nucleocapsid genes..."

Fig.  1A / L. 191-208. To demonstrate if the EVE-derived transcript is present, show that no PCR product is amplified when the RNA extract is used without cDNA synthesis. 

Fig. 3A. Include GenBank accession numbers of the virus sequences. It is not clear what black and grey highlights indicate, there is no consistency, many "black-highlighted" amino acids are not conserved. I suggest to present alignment with identical and homologous amino acid positions marked, for examples as in Clustal omega output (* : . )  

Fig, 3B. Include GenBank accession numbers for the virus sequences. Include bootstrap values.

Fig. 4E and 4F. The size distributions graph (Fig, 4E) suggest that NcP2-derived small RNA have both positive and negative polarities, but Fig. 4F shows only negative polarity. 

L. 290-291 (Fig. 6). It is indicated in Fig 6 that Phlebotomous samples (P. arabicus, P. argentipes. P. papatasi, P. sergent and P. schwetzi) are from Pragues, Central Europe. I am not sure if it these insect species live there ids the wild. Indicate where ethos samples were sourced originally. 

L.268-270  "...the density of small RNAs along the sequences were discontinuous, with hotspots at specific regions, and coverage concentrated in one strand (Figure 4)"  and L. 323-353.

How does the small RNA coverage for EV regions compared with that of other coding-non-coding region of the fly genome? 
